# Promoter-specific changes in initiation, elongation, and homeostasis of histone H3 acetylation during CBP/p300 inhibition

**Emily Hsu[1†], Nathan R Zemke[1‡], Arnold J Berk[1,2]\***

[1]Molecular Biology Institute, UCLA, Los Angeles, United States; [2]Department of Microbiology, UCLA, Los Angeles, United States

**\*For correspondence:**
arnold.berk@icloud.com

**Present address:** [†]Department of Biochemistry and Molecular Medicine and the Norris Comprehensive Cancer Center, Keck School of Medicine, University of Southern California, Los Angeles, United States; [‡]Department of Cellular and Molecular Medicine, University of California, San Diego School of Medicine, La Jolla, United States

**Competing interests:** The authors declare that no competing interests exist.

**Abstract** Regulation of RNA polymerase II (Pol2) elongation in the promoter-proximal region is an important and ubiquitous control point for gene expression in metazoans. We report that transcription of the adenovirus 5 E4 region is regulated during the release of paused Pol2 into productive elongation by recruitment of the super-elongation complex, dependent on promoter H3K18/27 acetylation by CBP/p300. We also establish that this is a general transcriptional regulatory mechanism that applies to ~7% of expressed protein-coding genes in primary human airway epithelial cells. We observed that a homeostatic mechanism maintains promoter, but not enhancer, H3K18/27ac in response to extensive inhibition of CBP/p300 acetyl transferase activity by the highly specific small molecule inhibitor A-485. Further, our results suggest a function for BRD4 association at enhancers in regulating paused Pol2 release at nearby promoters. Taken together, our results uncover the processes regulating transcriptional elongation by promoter region histone H3 acetylation and homeostatic maintenance of promoter, but not enhancer, H3K18/27ac in response to inhibition of CBP/p300 acetyl transferase activity.

## Introduction

In addition to RNA polymerase II (Pol2) pre-initiation complex (PIC) assembly and initiation, the transition from promoter-proximal paused Pol2 to productively elongating Pol2 is an essential step in gene transcription and an important process in the overall multicomponent orchestration of gene expression in metazoans (*Core et al., 2008*; *Seila et al., 2008*; *Core and Adelman, 2019*). After the recruitment of Pol2 to a promoter by its general transcription factors and assembly of a PIC (*Nogales et al., 2017*; *Sainsbury et al., 2015*), transcription initiation occurs concurrently with TFIIH phosphorylation of Ser5 of the Pol2 heptapeptide repeat C-terminal domain (CTD) (*Cramer, 2019*). In metazoan cells, Pol2 then transcribes approximately 30–60 bases downstream of the transcription start site (TSS) and then pauses because it is bound by negative elongation factor (NELF) and DRB-sensitivity-inducing factor (DSIF, Spt4 and Spt5 in *Saccharomyces cerevisiae*) (*Jonkers and Lis, 2015*; *Adelman and Lis, 2012*). Association of elongation factor P-TEFb, including its enzymatic subunits CDK9-cyclin T, results in phosphorylation of NELF, DSIF, and Ser2 of the Pol2 CTD, whereupon NELF dissociates and Pol2 is released and proceeds to productive elongation (*Cramer, 2019*; *Jonkers and Lis, 2015*; *Adelman and Lis, 2012*; *Vos et al., 2018*).

Histone acetylation is well known to contribute to a permissive chromatin state for Pol2 PIC assembly at active promoters, and there is recently published work concerning its function in facilitating transcriptional elongation as well. For example, the chromatin reader protein BRD4 is thought to recruit P-TEFb (CDK9-cyclin T) to promoters and serves as a Pol2 elongation factor dependent on its interactions with acetylated histone lysines through its bromodomains (*Kanno et al., 2014*). In addition, H3 acetylation mediated by the *Drosophila* CBP ortholog stimulates productive elongation

past the +1 nucleosome (*Boija et al., 2017*). Recruitment of the yeast histone chaperone FACT by acetylated H3 has also been shown to stimulate elongation (*Pathak et al., 2018*).

The super-elongation complex (SEC) is a multi-subunit complex comprising P-TEFb (CDK9-cyclin T) along with AF4/FMR2 proteins AFF1/4, ELL family members ELL1/2/3, ELL-associated factors EAF1/2, and one or the other highly homologous proteins AF9 or ENL containing YEATS acetyl-lysine-binding domains (*Luo et al., 2012*). There are various forms of the SEC, including SEC-like complexes that contain different combinations of elongation factors suggesting diversity in their regulatory mechanisms (*Luo et al., 2012*). The central serine/threonine-kinase P-TEFb (CDK9-cyclin T), an AFF scaffold protein, and ENL or AF9 are consistent components of SEC complexes. ENL and AF9 have been functionally linked to SEC recruitment to acetylated chromatin via their YEATS domains (*Li et al., 2014*; *Gates et al., 2017*). The SEC then stimulates transcription elongation through phosphorylation by CDK9 of NELF, DSIF, and Ser2 of the Pol2 CTD repeat, as well as interactions with the PAF1 complex (*He et al., 2011*), which blocks NELF-binding to Pol2 (*Cramer, 2019*), and DOT1L, which deposits the active chromatin modification H3K79me in the first intron (*Li et al., 2014*; *Huff et al., 2010*). Importantly, AF9 and ENL YEATS domains bind to active chromatin marks H3K9ac and H4K15ac (*Gates et al., 2017*), and, to a lesser extent, H3K18/27ac (*Li et al., 2014*), and are essential for SEC-dependent activation of a luciferase reporter driven by the HIV-1 LTR (*He et al., 2011*). Despite these conclusions, the function of histone acetylation during the transition from promoter-proximal paused to productively elongating Pol2 remains incompletely understood.

We previously reported that CBP/p300 acetylation of H3K18 and K27 in the two to three nucleosomes spanning the TSS had very different effects on distinct steps in transcription from different human adenovirus 5 (HAdV-5) early promoters (*Hsu et al., 2018*). At the E3 promoter, loss of H3K18/27ac in the promoter region had little effect on PIC assembly, and the rate of E3 mRNA synthesis was only modestly reduced (less than twofold) compared to transcription activated by wt E1A, which induces H3K18/27ac at the early viral promoters. In contrast, PIC assembly at the E2early promoter was almost eliminated by loss of promoter H3K18/27ac, and E2 mRNA synthesis was undetectable at 12 hr post-infection (p.i.) (*Hsu et al., 2018*). For E4, loss of promoter H3K18/27ac had little effect on PIC assembly but caused a significant (approximately tenfold) decrease in E4 transcription at 12 hr p.i. (*Hsu et al., 2018*). This result was particularly striking as it suggested that E4 transcription is regulated by promoter H3K18/27 acetylation at a step in transcription subsequent to PIC assembly, possibly during release of promoter-proximal paused Pol2.

To investigate the function of H3K18/27ac in transcriptional elongation at E4, we mapped the association of transcriptionally active Pol2 on the Ad5 genome using Global Run-On sequencing (GRO-seq) (*Core et al., 2008*). We found defective paused Pol2 release at E4 in cells expressing an E1A mutant ('E1A-DM') with polyalanine substituted for two highly acidic regions of the E1A activation domain (AD) that each mediate an interaction with CBP/p300 (*Hsu et al., 2018*). ChIP-seq for BRD4 and SEC components CDK9, AF9, and ENL revealed decreased SEC recruitment to E4 by E1A-DM compared to wt E1A. Using the specific small molecule inhibitor of CBP/p300 acetyl transferase activity A-485 (*Lasko et al., 2017*), we determined that CBP/p300 histone acetyl transferase (HAT) activities are essential for maximal paused Pol2 release at the E4 promoter, but not at the E3 promoter.

We then extended our studies to the human genome, where we found that 2 hr of A-485 treatment resulted in hypoacetylation of total cell H3K18/27 to a new, extensively hypoacetylated steady state. This was associated with defective pause release at a subset of active genes (~7%) where promoter H3K18/27ac was decreased by the drug. Differences in the sensitivity of transcription from different promoters to H3K18/27ac correlated with differences in SEC component association with the genes after A-485 treatment. This was similar to what we had observed for the HAdV-5 E4 promoter during activation by the multi-site E1A mutant (DM-E1A) with mutations in the E1A-AD acidic peptides required for CBP/p300 binding to the E1A-AD. We also found that at a subset of enhancers with greatly decreased H3K18/27ac in response to A-485 treatment, H3K9ac is sufficient for BRD4 binding and stimulation of Pol2 pause release. Based on these results, we propose mechanisms of BRD4 and SEC recruitment by histone H3 acetylation during the transition from promoter-proximal paused to productively elongating Pol2 and report a homeostatic process that maintains promoter H3K18/27ac.

# Results

CBP/p300 acetylation of promoter histone H3K18 and K27 stimulates paused Pol2 release at the HAdV-5 E4 promoter, but is not required at the E3 promoter.

Transcription from HAdV-5 early promoters is activated by the first viral proteins expressed following infection, the E1A isoforms, primarily large E1A (*Figure 1—figure supplement 1*). While transcriptional activation from the viral early promoters is entirely dependent on the interaction of the large E1A isoform with the mediator complex, transcription of E4 is stimulated an additional tenfold through interactions between CBP/p300 and two highly acidic regions immediately flanking the E1A mediator-binding region (large E1A aa residues 133—138 and 189—200; *Figure 1—figure supplement 1*; *Hsu et al., 2018*). Separate Ad5 expression vectors were constructed that express the wt E1A from the wt E1A promoter/enhancer or DM-E1A with several mutations that convert these acidic peptides in wt E1A to polyAla (*Figure 1—figure supplement 1b*).

To analyze the effects of promoter H3K18/27ac on Pol2 elongation through the early Ad5 genes, we applied the GRO-seq method, which reveals the position and direction of transcribing Pol2 by BrU-labeling of 3′-ends of nascent RNA transcripts in isolated nuclei (*Core et al., 2008*). The nuclei were first washed with the non-ionic detergent sarkosyl to remove proteins from chromatin that block transcription elongation and prevent Pol2 initiation, so only actively transcribing RNA polymerases at the time the nuclei were isolated produce BrU-labeled RNA (*Core et al., 2008*). To avoid possible effects of cellular mutations in stable cell lines, we performed these studies in primary human bronchial-tracheal epithelial cells (HBTECs) derived from human adult lung transplant donors. These HBTECs are a cell culture model for the airway epithelial cells infected by HAdV-5 in humans.

We infected HBTECs with the wt E1A vector or the DM-E1A vector expressing mutant E1A with polyAla substitutions of the two highly acidic peptides flanking CR3 (*Figure 1—figure supplement 1*). Wt E1A binds CBP/p300 through these highly acidic peptides, inducing histone H3 acetylation at K18 and K27 by the CBP/p300 acetyl transferase domain at the viral E2early, E3, and E4 promoter regions (*Hsu et al., 2018*). In cells expressing DM-E1A, which does not interact in vivo with CBP/p300 through the E1A activation domain (E1A-AD) (*Hsu et al., 2018*), there was far less H3K18/27ac at these viral early promoter regions (*Figure 1a*). To express equal steady-state levels of the wt E1A and less stable DM-E1A proteins, infections were performed at a twentyfold higher multiplicity of infection for the DM-E1A vector than for the wt E1A vector (*Hsu et al., 2018*; *Figure 1—figure supplement 1*). Cells infected with the wt E1A vector were also coinfected with sufficient E1A deletion mutant *dl312* to maintain the same number of viral DNA molecule templates for the viral early regions (~100 viral DNA molecules per nucleus) in cells expressing the same level of wt E1A and the DM-E1A protein (*Hsu et al., 2018*; *Figure 1—figure supplement 1*).

GRO-seq data at 12 hr p.i. with the vector expressing wt E1A revealed peaks of paused Pol2 with the expected orientation and location of promoter-proximal paused Pol2, ~40–60 bp downstream from the E3 and E4 TSSs (*Figure 1a*, highlighted, *Figure 2a*). At 12 hr p.i., very low GRO-seq signal was observed at the E2 early promoter or within the E2 gene body in wt E1A expressing cells compared to E3 and E4 (*Figure 1a*). This is probably because E2early transcription is delayed compared to E3 and E4 in these primary cells and increases by 18 hr p.i. (*Hsu et al., 2018*). The low GRO-seq signal in the E2early promoter region and gene body was decreased further in cells expressing DM-E1A compared to wt E1A (*Figure 1a*), supporting our previous conclusion that E2early transcription is regulated by H3K18/27ac in the promoter region because it is required for rapid PIC assembly (*Hsu et al., 2018*).

To determine the degree of promoter-proximal pausing in the E2, E3, and E4 promoter regions where Pol2 association is detected by ChIP-seq at 12 hr p.i. (*Hsu et al., 2018*), we calculated the Pol2 pausing index (PI, *Core et al., 2008*). The PI equals the number of GRO-seq reads in the promoter-proximal region (TSS to +200 bp) divided by the total GRO-seq reads in the gene body (+201 to TTS). The GRO-seq reads in the promoter-proximal region reflect the amount of promoter-proximal paused Pol2 in the population of cells at the time the nuclei were isolated, while the GRO-seq reads in the gene body reflect the amount of elongating Pol2 subsequent to pause release. Therefore, an increase in PI indicates a reduced rate of promoter-proximal pause release.

After activation by DM-E1A, the PI at E4 increased 2.5-fold on average in three biological replicate GRO-seq experiments, a significant increase compared to the degree of pausing at E4 when activated by wt E1A (*Figure 1b*). Because the vectors expressing wt E1A and DM-E1A had different

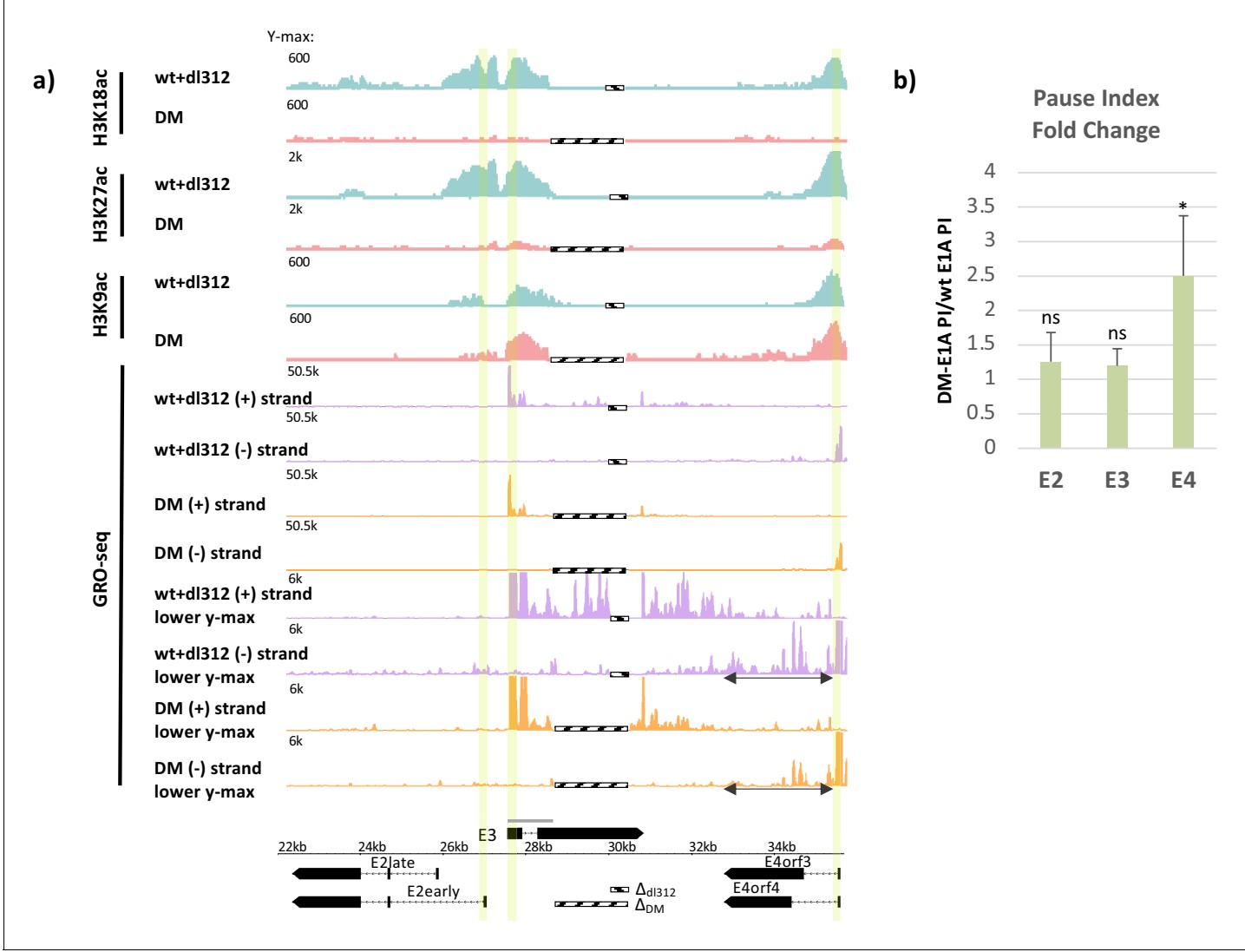

**Figure 1.** Promoter H3K18/27 acetylation activated by E1A activation domain (E1A-AD)-CBP/p300 interaction stimulates paused polymerase II (Pol2) release at adenovirus promoter E4. (**a**, bottom) Map of the major human adenovirus 5 early E2, E3, and E4 mRNAs. Deletions in the E3 regions of *dl*312 and the E1A-DM vector are shown by cross-hatched horizontal bars. Shared E3 sequence used as the E3 gene body is indicated by the gray bar above E3. Vertical stripes highlighted in yellow indicate promoter-proximal regions. Global Run-On sequencing (GRO-seq) counts from primary human bronchial-tracheal epithelial cells infected with wt+*dl*312 or DM vectors at 12 hr post-infection were plotted on the Ad5 genome with H3K18ac, H3K27ac, and H3K9ac ChIP-seq data (*Hsu et al., 2018*). GRO-seq tracks are shown for the two viral DNA strands (+, transcribed to the right; and –, transcribed to the left), with two different y-axis scales to allow visualization of high- and low-amplitude peaks. The double-headed arrows in the GRO-seq plots in the E4 region refer to gene body regions discussed in the text. (**b**) Average fold change in pause index for E2early, E3, and E4 in cells expressing DM-E1A compared to wt E1A. Pause index is the ratio of reads in the promoter region (transcription start site [TSS] to +200) to reads in the gene body (+200 to TTS). Error bars represent standard deviation of three biological replicates. Paired t-test comparing wt E1A and DM-E1A for E2, E3, and E4. * indicates a significant difference (p-value<0.05) between cells expressing wt E1A and DM-E1A. 'ns' indicates no significant difference.

The online version of this article includes the following source data and figure supplement(s) for figure 1:

**Source data 1.** qRT-PCR for E2early, E3, and E4 pre-mRNA transcripts in cells treated with DMSO or A-485.

**Figure supplement 1.** Small and large E1A protein interaction and E1A mutants.

size deletions in E3 due to differences in their constructions (*Figure 1a*, bottom), the calculation of PI for E3 was based on the ~1.1 kb region of E3 common to both vectors (gray line above E3 in the Ad5 map, *Figure 1a*, bottom). In contrast to E4, there was no significant change in PI at E3 where promoter H3K18/27 acetylation had only a modest effect on transcription (*Hsu et al., 2018*; *Figure 1b*). Similar to E3, the low level of GRO-seq counts at the E2early promoter region showed

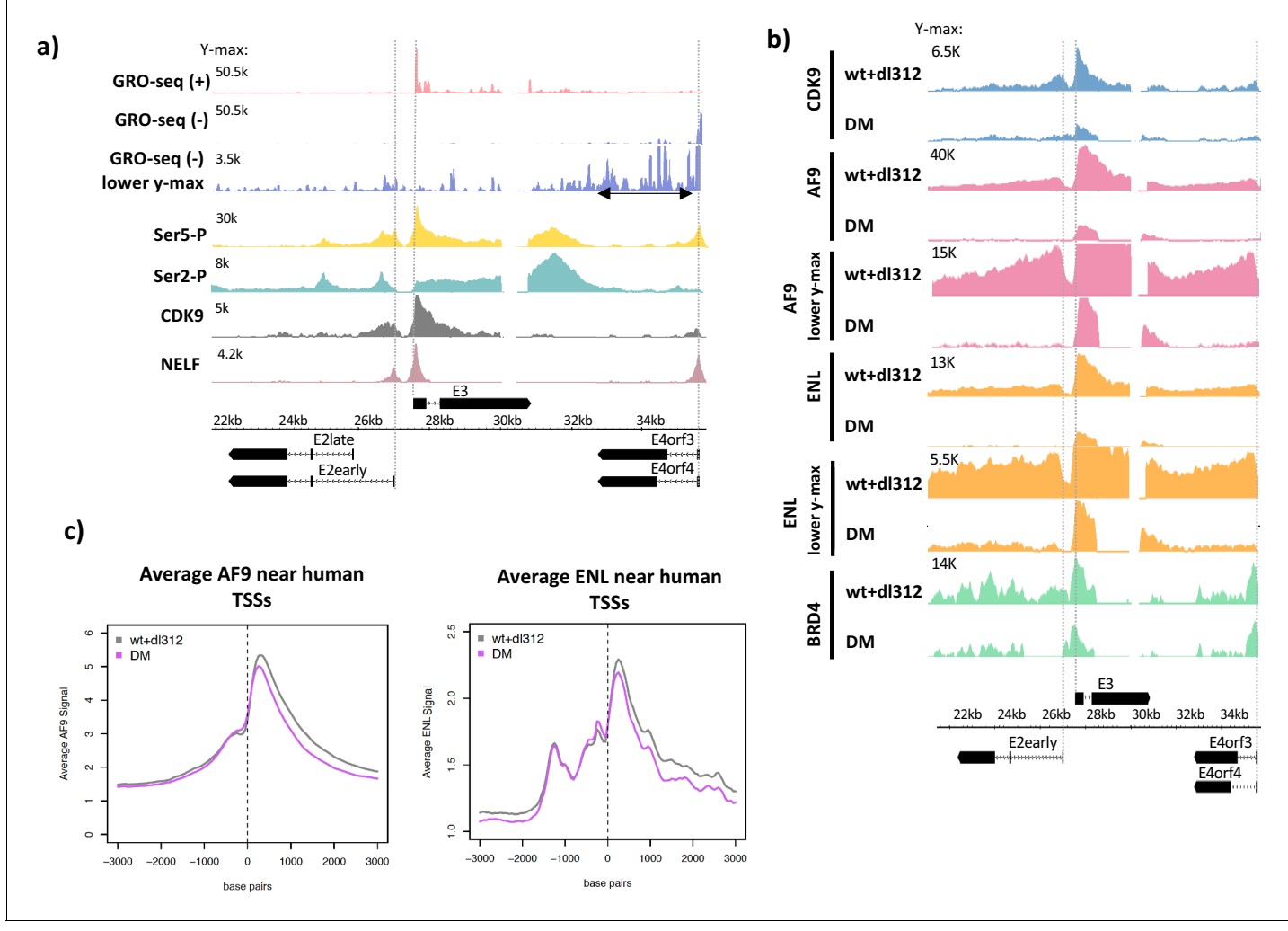

**Figure 2.** Ser5-P, Ser2-P, CDK9, negative elongation factor (NELF), and super-elongation complex (SEC) subunits on the Ad5 genome. (a, top) Global Run-On sequencing in cells expressing wt E1A; (bottom) ChIP-seq of C-terminal domain (CTD) Ser5-P, CTD Ser2-P, CDK9, and NELF. (b) CDK9, AF9, ENL, and BRD4 ChIP-seq in cells expressing wt or DM E1A. AF9 and ENL ChIP-seq data are plotted with two different y-axes. (c) Average plots of AF9 and ENL binding near transcription start sites on the human genome.

an insignificant change in PI between wt E1A and DM E1A (*Figure 1b*), suggesting that H3K18/27ac acetylation does not affect pause release at the E2early promoter and primarily promotes Pol2 initiation. Therefore, loss of H3K18/27 promoter acetylation produced a smaller defect on promoter-proximal Pol2 pause release at the E2 and E3 promoters than at the E4 promoter.

## Decreased Pol2 pause release in the E4 promoter-proximal region correlated with decreased association of SEC subunits CDK9, AF9, and ENL

The CDK7 subunit of TFIIH phosphorylates Ser5 of the Pol2 CTD repeat during transcription initiation, and subsequently the CDK9 subunit of P-TEFb phosphorylates Ser2 of the CTD repeat, NELF, and DSIF causing release of Pol2 arrested by NELF binding in the promoter-proximal region, and the transition to productive elongation (*Cramer, 2019*; *Jonkers and Lis, 2015*; *Rahl et al., 2010*; *Yamada et al., 2006*). To characterize these mechanisms on the HAdV-5 genome, we performed ChIP-seq for Pol2 Ser5-P, Pol2 Ser2-P, NELF, and CDK9 in cells expressing wt E1A (*Figure 2a*). At

E2early, Ser5-P peaked near the TSS and decreased throughout the gene body, a distribution that is typical in yeast that also has short genes with few introns (*Buratowski, 2009*), as well as mouse ES cells (*Rahl et al., 2010*) with the much longer, multi-exon, long-intron genes typical of vertebrates. We observed two Ser2-P peaks in the E2early gene body, one positioned just downstream of the TSS probably corresponding to paused Pol2 in complex with NELF, and a second peak upstream of the E2early second exon. A small Ser5-P peak was also observed at this position (*Figure 2a*). These Pol2 peaks just upstream of the E2 second exon may be explained by a previously reported reduction in elongation rate over some exons, proposed to influence splice site recognition and spliceosome assembly (*Jonkers et al., 2014*; *Martin et al., 2013*).

Both CDK9 and NELF peaks occurred at the expected E2early, E3, and E4 pause sites ~40 bp downstream of the TSSs (*Figure 2a*). Broad enrichment of Ser2-P and Ser5-P Pol2 was also observed downstream of the E3 and E4 poly(A) sites. Increased Pol2 Ser2-P and Ser5-P downstream from cellular poly(A) sites is observed at most genes in mammalian cells and is thought to result from a decrease in Pol2 elongation rate following nascent RNA cleavage at the poly(A) site (*Rahl et al., 2010*).

We next asked if defective paused Pol2 release after activation by DM-E1A was due to decreased recruitment of P-TEFb-containing complexes. A large percentage of P-TEFb exists in complex with the 7SK snRNP where its CDK9 kinase activity is inhibited and it is sequestered from chromatin (*Nguyen et al., 2001*; *Yang et al., 2001*; *Li et al., 2005*). Release of P-TEFb from the 7SK snRNP enables its integration into complexes with activated CDK9 kinase activity, including the SEC and a complex comprising P-TEFb and BRD4 (*Chen et al., 2018*, *Jang et al., 2005*). Integration into these complexes allows active CDK9 to be targeted to promoters and enhancers where it phosphorylates its targets and stimulates paused Pol2 release (*Kanno et al., 2014*). To determine the effects of H3K18/27ac on SEC and P-TEFb-BRD4 recruitment to early adenovirus genes, we performed ChIP-seq for CDK9, AF9, ENL, and BRD4 on the HAdV-5 genome in infected cells (*Figure 2b*). Reduced H3K18/27ac in DM-E1A vector-infected cells compared to wt E1A-expressing cells correlated with decreased CDK9, AF9, and ENL association with the viral early promoters and gene bodies compared to cells expressing wt E1A (*Figure 2b*). Importantly, we did not observe decreases in average AF9 and ENL association with TSSs of most human genes in the same infected cells expressing DM-E1A, demonstrating the specificity of this effect on SEC subunit association at the viral early promoters (*Figure 2c*).

The ChIP-seq counts for AF9 and ENL in cells expressing wt E1A were twofold to fourfold higher at the E3 promoter compared to the E2early and E4 promoters. In cells expressing the DM-E1A, which caused hypoacetylation of H3K18/27 at the early viral promoters, the ChIP-seq signals for AF9 and ENL at the E2early and E4 promoters were virtually eliminated, but clear peaks for AF9 and ENL persisted at the E3 promoter region, although they were reduced considerably compared to cells expressing wt E1A (*Figure 1*, *Figure 2b*; *Hsu et al., 2018*). BRD4 association at the E2early, E3, and E4 TSSs changed very little in cells infected with the DM-E1A vector compared to the wt E1A vector, although it was reduced to about 50% the level with wt E1A within the transcription units (*Figure 2b*). These data suggest that association of the wt E1A-AD with CBP/p300 and the resulting acetylation of H3K18/27 in the early promoter regions are required for maximal recruitment of the SEC to the viral early promoters.

## CBP/p300 acetyl transferase activity is required for efficient Pol2 pause release and recruitment of AF9, ENL, and BRD4 at E4

A-485 is a potent and specific small molecule inhibitor of CBP/p300 acetyl transferase activity that competes with acetyl-CoA for binding to the acetyl transferase domain active site (*Lasko et al., 2017*). Decreased total cell H3K18ac after A-485 treatment in HBTECs was confirmed by western blot (*Figure 3a*). ChIP-seq for H3K9ac, H3K18ac, and H3K27ac in wt E1A vector-infected cells treated with A-485 for 2 hr demonstrated A-485 inhibition of H3K18/27ac, as expected (*Lasko et al., 2017*; *Weinert et al., 2018*), while H3K9ac was modestly affected at viral early promoters (*Figure 3b*). As a measure of the transcription rate of the early viral genes, we assayed pre-mRNA levels using qRT-PCR of intronic RNA isolated from HBTECs expressing wt E1A treated with 10 µM A-485 or control DMSO (dimethyl sulfoxide) vehicle alone for 2 hr. We observed significant decreases in E2early and E4 pre-mRNA after A-485 treatment, while E3 pre-mRNA was not

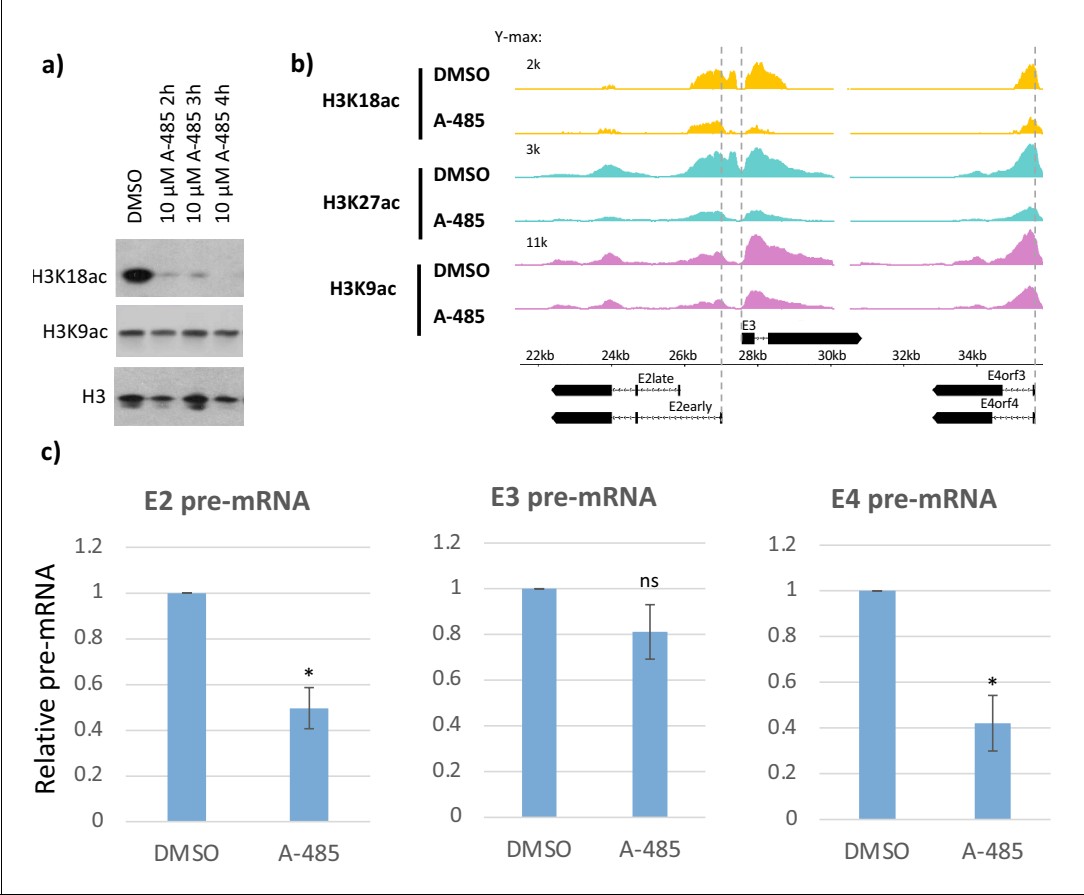

**Figure 3.** CBP/p300 histone acetyl transferase inhibitor A-485 causes H3 hypoacetylation and decreased early viral gene expression. (a) Western blot for H3K18ac, H3K9ac, and total H3 in human bronchial-tracheal epithelial cells treated with control vehicle DMSO (dimethyl sulfoxide) or 10 µM A-485 after 2, 3, or 4 hr. (b) H3K18ac, H3K27ac, and H3K9ac ChIP-seq at early viral promoters in cells treated with control vehicle DMSO (dimethyl sulfoxide) or 10 µM A-485 for 2 hr. (c) qRT-PCR for E2early, E3, and E4 pre-mRNA transcripts in cells treated with DMSO (dimethyl sulfoxide) or 10 µM A-485 for 2 hr. Averages and standard deviations shown for three successive experiments. Paired t-test, *p<0.05 between cells treated with DMSO (dimethyl sulfoxide) and A-485. 'ns' indicates no significant difference between cells treated with DMSO (dimethyl sulfoxide) and A-485.

The online version of this article includes the following source data for figure 3:

**Source data 1.** Average fold change in pause index for E2early, E3, and E4 in cells expressing wt E1A compared to DM E1A.

decreased significantly (*Figure 3c*). This result confirms that E3 transcription is less dependent on promoter H3K18/27ac than E2early and E4 transcription (*Hsu et al., 2018*).

To determine if A-485-induced H3K18/27 hypoacetylation correlates with changes in elongation, we performed GRO-seq in DMSO or A-485-treated cells and calculated the pausing indices for transcription of the viral early genes (*Figure 4a, b*). There was little difference in PI with A-485 treatment at the E2early (DMSO (dimethyl sulfoxide) PI = 0.13, A-485 PI = 0.12) or E3 (DMSO (dimethyl sulfoxide) PI = 1.20, A-485 PI = 1.19) promoters, but a clear increase in PI was observed for E4 (DMSO (dimethyl sulfoxide) PI = 0.79, A-485 PI = 1.13) (*Figure 4b*). These results indicate that release of paused Pol2 in the E4 promoter-proximal region depends on CBP/p300 HAT activity and is consistent with a requirement for promoter H3K18/27 acetylation for maximal E4 elongation.

To determine if inhibition of CBP/p300 HAT activity resulted in defective SEC recruitment at E4, we performed AF9, ENL, and BRD4 ChIP-seq in cells infected with the wt E1A vector after DMSO (dimethyl sulfoxide) or A-485 treatment. As with cells expressing DM-E1A, A-485 treatment greatly decreased AF9 association at the E2early, E3, and E4 promoter regions, and decreased total

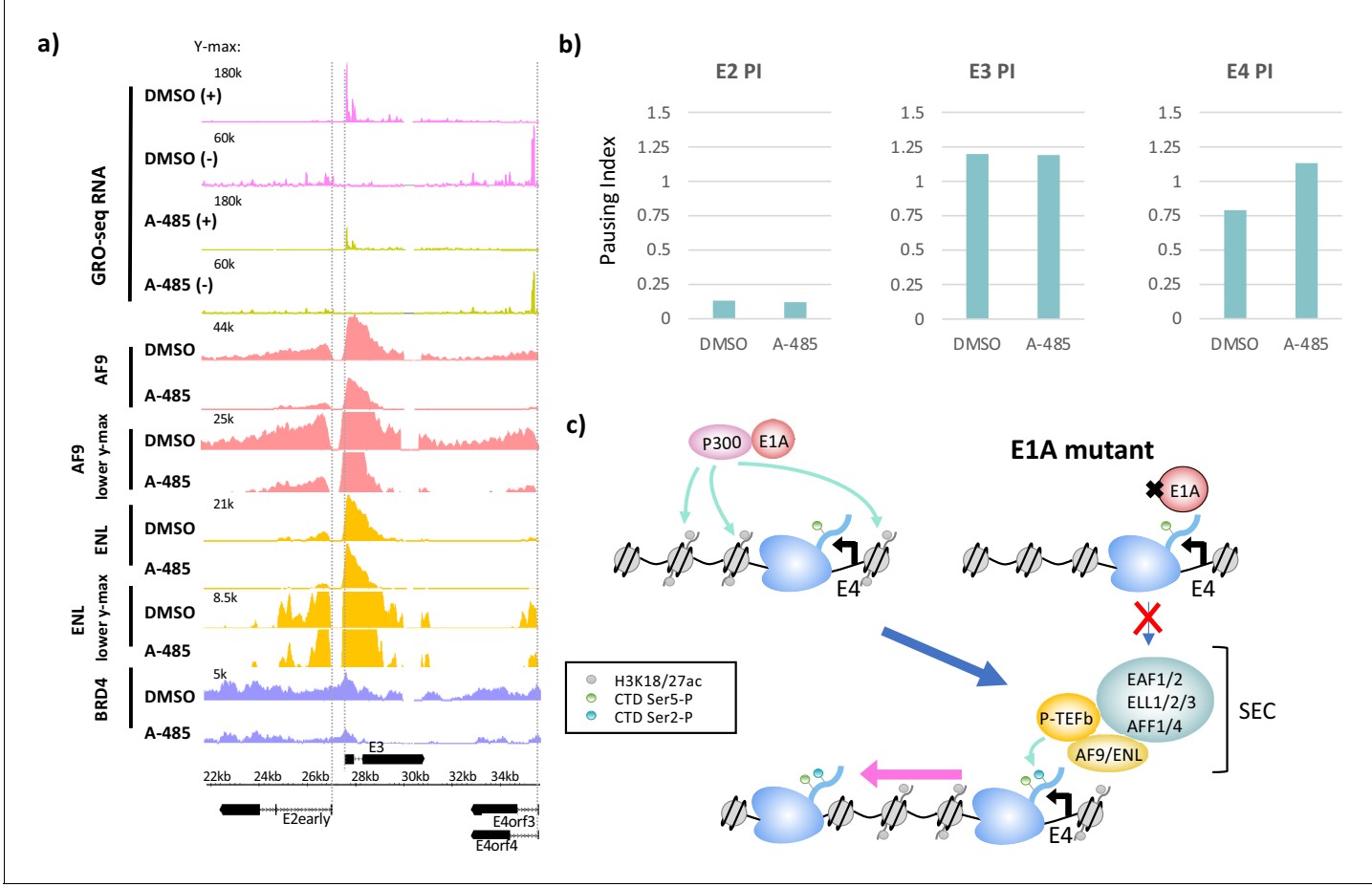

**Figure 4.** CBP/p300 histone acetyl transferase inhibition by A-485 results in defective polymerase II pause release and decreased super-elongation complex (SEC) and BRD4 association at E4. (a) Global Run-On sequencing (GRO-seq) in cells expressing wt E1A treated with DMSO (dimethyl sulfoxide) or 10 μM A-485 for 2 hr. GRO-seq tracks are plotted with ChIP-seq for AF9, ENL, and BRD4 in cells treated with DMSO (dimethyl sulfoxide) or 10 μM A-485 for 2 hr. Both AF9 and ENL ChIP-seq tracks are shown with two different y-axes. (b) Pause indexes for E2early, E3, and E4 in cells treated with DMSO (dimethyl sulfoxide) versus A-485. (c) Model for regulation of E4 elongation by SEC recognition of CBP/p300-E1A-mediated H3K18/27ac. The online version of this article includes the following source data for figure 4:

**Source data 1.** Pause indexes for E2early, E3, and E4 in cells treated with DMSO compared to A-485 treated cells.

ENL ChIP-seq counts from the E2early TTS to the right end of the genome from 13,712,875 to 10,804,279 reads, an ~20% decrease (*Figure 4a*). We also observed decreased BRD4 throughout the transcribed early regions in A-485-treated cells. Taken together, these data show that maximal recruitment of SEC complexes to viral early promoters requires H3K18/27ac by the CBP/p300 acetyl transferase catalytic domain targeted to the promoter region by the interaction between CBP/p300 and the E1A-AD acidic regions (*Figure 4c*).

## CBP/p300 HAT inhibition by A-485 affects H3 acetylation of cellular chromatin differently at promoters and enhancers

To determine if the effects of H3K18/27ac on Pol2 pause release at E4 is a general mechanism that also applies to transcription of cellular genes, we shifted our study to the human genome. First, we characterized the changes in H3K18/27ac in HBTECs after 2 hr treatment with 10 μM A-485. Western blotting demonstrated an extensive decrease in total cellular H3K18ac, which approached steady state by 2 hr after addition of A-485, as expected (*Lasko et al., 2017*; *Weinert et al., 2018*; *Figure 3a*). H3K18ac and H3K27ac enriched genomic regions were determined by separate ChIP-seq analyses with antibodies specific for either H3K18ac or H3K27ac. H3K18/27ac are enriched at

both active promoters and enhancers. Enhancers can be divided into typical enhancers and super-enhancers, which are dense clusters of neighboring enhancers that often activate highly expressed cell identity genes (*Whyte et al., 2013*). We used the ranking of super-enhancers algorithm (*Whyte et al., 2013*; *Lovén et al., 2013*; *Hnisz et al., 2013*) with H3K27ac ChIP-seq data to determine super-enhancer regions in HBTECs (*Zemke et al., 2019*). H3K27ac peaks >2.5 kb from the nearest TSS but outside of super-enhancers were classified as typical enhancers. Comparing the average signals for H3K18ac and H3K27ac, we observed the expected decreases due to A-485 treatment at both typical and super-enhancers (*Figure 5a*). But, on average, A-485 caused greater decreases in both H3K18ac and H3K27ac at super-enhancers than typical enhancers (*Figure 5a*). This pattern was also true for H3K9ac, although the hypoacetylation by A-485 was to a much lesser extent (*Figure 5—figure supplement 1*).

An unexpected result was the *increase* in the average level of H3K18ac and H3K27ac at all TSSs in cells treated with A-485 (*Figure 5a*). Furthermore, the percentage of total H3K27ac and H3K18ac peaks increased at promoters and decreased at enhancers after A-485 treatment, while the effects on H3K9ac peaks were minimal (*Figure 5—figure supplement 2*). These observations indicate that homeostatic mechanisms function to maintain H3K18/27ac at promoters when CBP/p300, the principal cellular acetyl transferases for these sites (*Weinert et al., 2018*; *Horwitz et al., 2008*; *Ferrari et al., 2014*), are extensively inhibited.

## A-485 affects cellular genes during both transcriptional initiation and elongation

We were also curious about whether A-485 treatment affected transcription of human genes during both initiation and elongation and whether or not there are variations in the effects of A-485 at different human promoters, as observed on the HAdV-5 genome. GRO-seq reads from control DMSO (dimethyl sulfoxide) and A-485-treated cells were aligned to the human genome to determine the fraction of genes affected by A-485 at different stages in transcription. We limited our analysis to protein-coding genes with active promoters containing at least 20 GRO-seq counts in the promoter region (TSS to +200) and a significant promoter H3K9ac peak (q-value <0.05). Out of 10,062 such active protein-coding genes, we found 686 (6.8%) with defective pause release after A-485 treatment (greater than twofold increase in PI, referred to as '2XPI' genes). In total, 885 genes (8.8% of active genes) were inhibited for initiation (<50% the GRO-seq counts in the promoter region [TSS – 200 bp] compared to control DMSO-treated cells). We refer to these genes as '2XIn' genes for twofold decreased initiation from A-485. Also, 217 genes passed the criteria for both groups, indicating that both transcription initiation and promoter-proximal pause release were reduced by A-485 treatment for this small set of genes (~2.15% of active genes, *Figure 5b*). Gene ontology analysis of 2XPI genes (*Figure 5—figure supplement 3*) revealed enrichment of genes involved in the development of multiple tissues (*Figure 5—figure supplement 3*), which is typical of genes regulated by enhancers (*Bulger and Groudine, 2011*; *Levine, 2010*). We also performed motif analysis for transcription factor binding sites in promoter regions of the 2XPI genes, revealing enrichment for several transcription factors involved in developmental regulation including serum response factor known to regulate angiogenesis as well as neural development and to be a key regulator of mammalian mesoderm formation (*Arsenian et al., 1998*; *Schwartz et al., 2014*; *Franco and Li, 2009*; *Knöll and Nordheim, 2009*). For 2XIn genes, there was enrichment of binding motifs associated with NKX6-3, which functions in the development of various tissues, most notably those in the central nervous system and gastrointestinal tissues (*Yoon et al., 2015*).

Since we observed that super-enhancers and transcription of genes involved in development are particularly sensitive to A-485, we asked if 2XPI and 2XIn genes are more likely than most genes to be associated with super-enhancers. We found that, indeed, both 2XPI and 2XIn genes were significantly closer to super-enhancers than randomly selected genes (*Figure 5c*). This suggests that super-enhancer-associated genes are more dependent on H3K18/K27ac for transcription initiation and elongation than the average expressed gene.

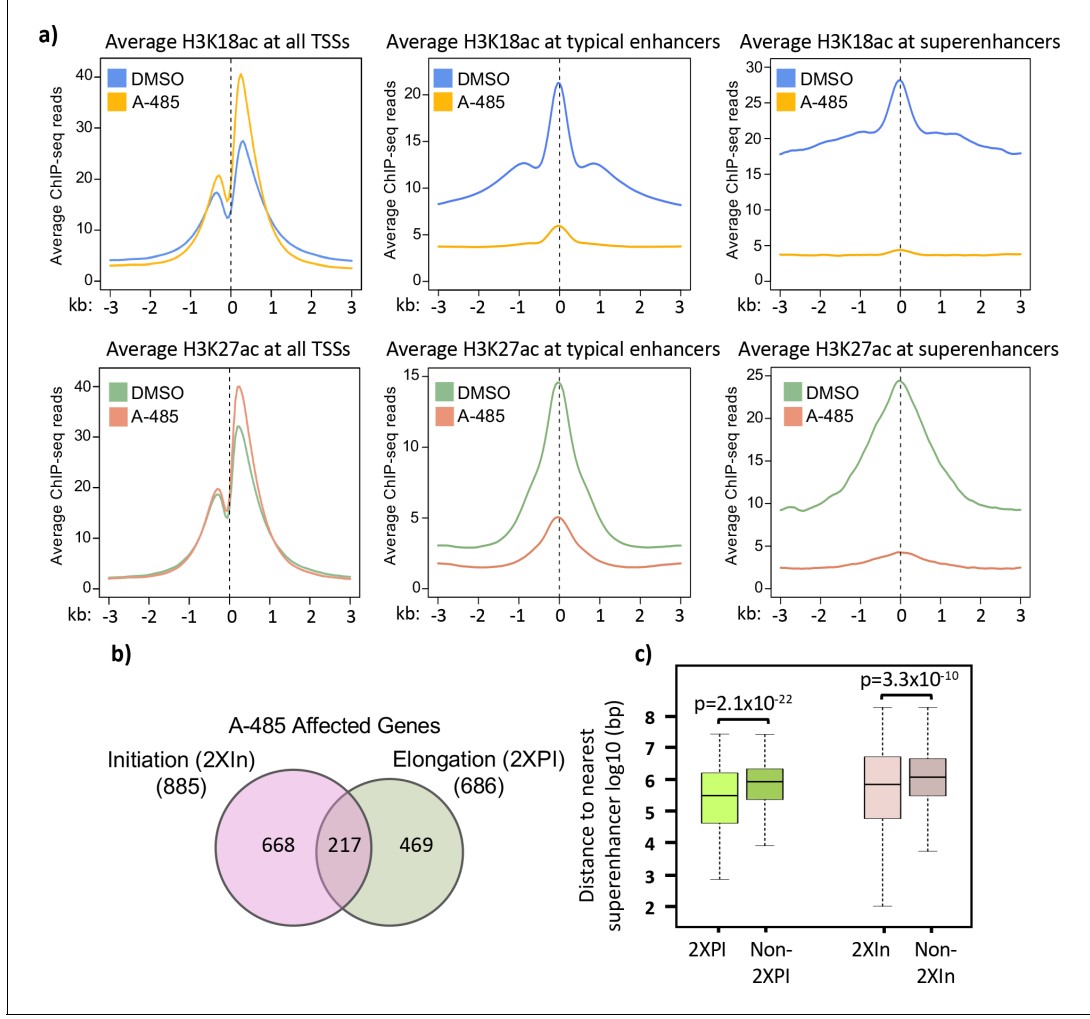

**Figure 5.** Treatment with A-485 causes different effects on H3K18/27ac at promoters and enhancers and results in defects in both initiation and elongation. (a) Plots of average H3K18ac and H3K27ac signals at all human transcription start sites (TSSs), typical enhancers, and at super-enhancers in primary cells treated with A-485 or DMSO (dimethyl sulfoxide). (b) Venn diagram showing number of protein-coding genes with defects during transcription by A-485 during initiation (greater than twofold decrease in counts in the promoter region) and elongation (greater than twofold increase in pausing index). (c) Boxplots showing distributions of distances (log10 bp) from the nearest super-enhancer to the TSS for 2XPI genes and 2XIn genes versus the same numbers of genes without defects in transcription after A-485. p-Values are from two-sided t-test.

The online version of this article includes the following source data and figure supplement(s) for figure 5:

**Source data 1.** Lists of all active genes, 2XPI, and 2XIn genes.
**Figure supplement 1.** Average H3K9ac and BRD4 peaks at typical and super-enhancers and TATA box motifs in promoter regions.
**Figure supplement 2.** Distribution of H3K27ac, H3K18ac, and H3K9ac, and BRD4 peaks at promoters, typical enhancers, and super-enhancers.
**Figure supplement 2—source data 1.** Percentage of H3K27ac, H3K18ac, H3K9ac, and BRD4 peaks called in indicated annotated regions from DMSO versus A-485 treated cells.
**Figure supplement 2—source data 2.** Total number of H3K27ac, H3K18ac, H3K9ac, and BRD4 peaks in cells treated with DMSO versus A-485 treated cells as observed by ChIP-seq.
**Figure supplement 3.** Promoter motifs and gene ontologies of 2XPI and 2XIn genes.
**Figure supplement 3—source data 1.** Promoter proximal de novo motifs and gene ontology of 2XPI and 2XIn genes.

## A-485-sensitive Pol2 pause release and SEC recruitment at genes with low promoter region H3 acetylation

We plotted the average H3K18ac, K27ac, and K9ac ChIP-seq counts ±5 kb from the TSS for all active genes, 2XPI genes, and 2XIn genes (*Figure 6a*). H3K18ac and K27ac at TSSs for 2XPI and 2XIn genes decreased in response to A-485, as expected for a specific competitive inhibitor of the CBP/p300

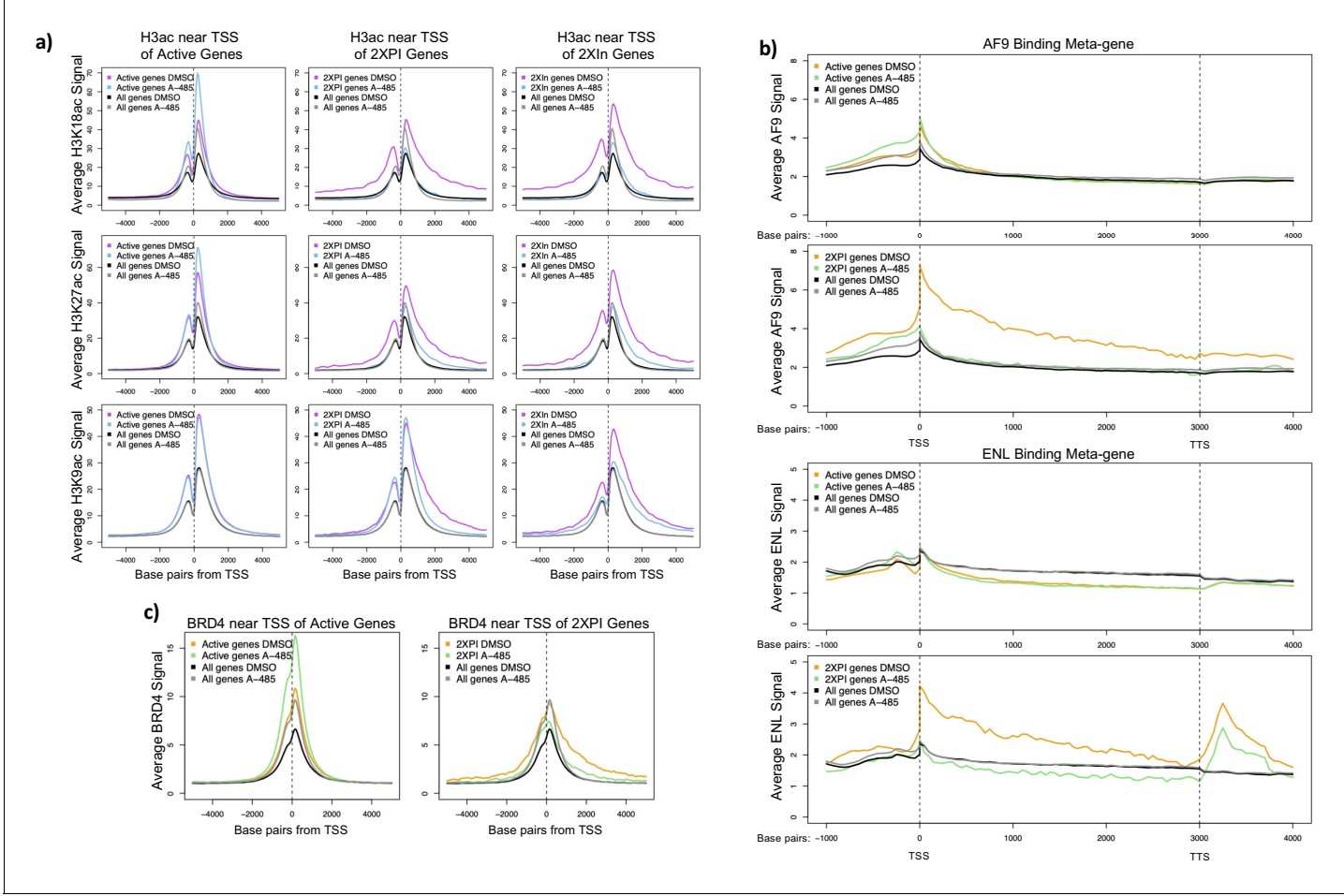

**Figure 6.** A-485-sensitive promoter H3K18/27ac and high super-elongation complex association at 2XPI genes. (**a**) Average H3K18ac, H3K27ac, and H3K9ac near transcription start sites (TSSs) for all active genes, 2XPI genes, and 2Xin genes after DMSO (dimethyl sulfoxide) or A-485 treatment. (**b**) Metagene plots of AF9 and ENL ChIP-seq counts across all active genes and 2XPI genes after DMSO (dimethyl sulfoxide) or A-485 treatment. (**c**) Average BRD4 near TSSs for all active genes and 2XPI genes after DMSO (dimethyl sulfoxide) or A-485 treatment.

The online version of this article includes the following figure supplement(s) for figure 6:

**Figure supplement 1.** Average AF9, ENL, and BRD4 binding in 2Xin genes in cells treated with DMSO or A-485.

acetyl transferase. But this was in contrast to the surprising *increase* in H3K18 and K27 acetylation on average near the TSSs for all active genes in response to A-485. Thus H3K18/27 acetylation in the promoter regions of 2XPI and 2Xin genes was decreased by A-485 as it is throughout the vast majority of the genome (*Figure 3a*), whereas the average H3K18/27 acetylation in the promoter regions for all active genes was increased by treatment with A-485 (*Figure 6a*). In contrast, H3K9ac did not change at the TSS in the average plot for all active genes or 2XPI genes after A-485 treatment, but did decrease for 2Xin genes (*Figure 6a*). This suggests that H3K9ac may be sufficient for initiation but not elongation of genes with promoter H3K18/27 hypoacetylation.

To determine if these decreases in H3 acetylation at 2XPI genes were correlated with decreased SEC component binding, we plotted metagene profiles of AF9 and ENL ChIP-seq counts for all active genes and 2XPI genes after DMSO- or A-485 treatment (*Figure 6b*). Remarkably, AF9 and ENL were extensively enriched at TSSs and throughout gene bodies of 2XPI genes compared to the average for all active genes. Further, after A-485 treatment, AF9 and ENL association with 2XPI genes fell to the average level for all genes, which did not change with A-485 treatment. Thus, genes with an increase in PI after A-485 treatment (2XPI genes) had a very high association of SEC complexes throughout their transcription units that was selectively sensitive to A-485 treatment.

Similarly, we found enrichment of AF9 and ENL at TSSs and throughout gene bodies in 2XIn genes, which fell to the level of the average for all genes after A-485 treatment (*Figure 6c*, *Figure 6—figure supplement 1*). This decrease in elongation factor association with 2XIn genes is likely explained in part because this group of genes was selected for having decreased Pol2 in the gene body with which elongation factors associate (*Figure 6—figure supplement 1*). BRD4 association at the TSSs of all active genes was increased with A-485, while association near the TSS of 2XPI and 2XIn genes was reduced by A-485, but to a far less extent than the decrease in AF9 and ENL.

### H3K9ac is sufficient for enhancer association with BRD4 that stimulates pause release at nearby genes

It was evident that BRD4 peaks were enriched at enhancers. Fifty-six percent of identified BRD4 peaks (see Materials and methods) were >2.5 kb from the nearest TSS in control cells treated with DMSO (dimethyl sulfoxide). Of these distal peaks, 84% overlapped with peaks of H3K27ac, indicating that these peaks were primarily at enhancers. We subsequently clustered all BRD4 peaks >2.5 kb from a TSS (i.e., within an enhancer) based on whether BRD4 association decreased to <50% of DMSO (dimethyl sulfoxide) control after A-485 treatment for 2 hr. There were 7554 peaks where BRD4 decreased to <50% of control after A-485 treatment and 6324 peaks where BRD4 association remained unaltered or was not reduced to <50% of control (referred to as A-485-sensitive and -resistant BRD4 enhancer peaks, respectively; *Figure 7*). At both A-485-resistant and -sensitive BRD4 enhancer peaks, H3K18/27ac decreased following A-485 treatment (*Figure 7a*). In contrast, H3K9ac decreased only at the A-485-sensitive BRD4 peaks but not at A-485-resistant BRD4 peaks. These observations indicate that enhancer H3K9ac correlates with BRD4 association and suggest that H3K9ac is sufficient for BRD4 association at enhancers.

We next asked if BRD4 enhancer association correlated with the extent of Pol2 pause release in the promoter-proximal region of nearby genes. For example, *COPS8* has enhancers downstream from the TSS in the first intron and beyond its TTS as indicated by H3K18/27ac peaks in control DMSO-treated cells (*Figure 7b*). H3K18/27ac, H3K9ac, and BRD4 association with these enhancer peaks was A-485 sensitive, but not their associations near the TSS (*Figure 7b*). A-485 caused only a modest decrease in the GRO-seq reads at the Pol2 pause site (~30%, *Figure 7b*) and, therefore, only an ~30% decrease in the amount of Pol2 that had initiated transcription at the *COPS8* TSS in this population of cells compared to control DMSO-treated cells. But A-485 caused a much greater decrease in GRO-seq reads relative to control DMSO-treated cells downstream from the *COPS8* promoter-proximal pause site, indicating that A-485 reduced the rate of Pol2 pause release for this gene (*Figure 7b*, *Figure 7—figure supplement 1*). Additionally, *NDRG1* and *PAG1* are examples of other genes with nearby enhancers that have A-485-sensitive BRD4 association (*Figure 7—figure supplements 2* and *3*). Similar to *COPS8*, A-485 treatment caused a decrease in release of promoter-proximal paused Pol2, but little decrease in Pol2 initiation at these genes (*Figure 7—figure supplements 2* and *3*). These results indicate that genes with A-485-sensitive BRD4 enhancer association are transcriptionally inhibited by A-485 during elongation.

In contrast to *COPS8, NDRG1*, and *PAG1, PTPRF* is a gene with A-485-resistant BRD4 association at nearby enhancer regions within its introns (*Figure 7b*, *Figure 7—figure supplement 1*). A-485 treatment reduced H3K18/27ac but not H3K9ac or BRD4 association at these enhancers. GRO-seq reads were reduced to approximately the same degree at the pause site and throughout the gene body, indicating regulation mainly at the initiation step. *CSF3* is also an example of a gene where A-485 greatly inhibited Pol2 initiation (*Figure 7—figure supplement 4*). We tested if the correlation between BRD4 enhancer association and efficient Pol2 pause release in the promoter-proximal region was observed broadly. To examine this, we compared the change in PIs after A-485 treatment for genes near A-485-sensitive BRD4 enhancer peaks with that of all active genes. We observed a significant increase in PI distribution for genes near A-485-sensitive BRD4 acting at distal enhancers to stimulate transcription elongation of target genes (*Kanno et al., 2014*).

## Discussion

There is a substantial base of knowledge establishing a correlation between histone N-terminal tail lysine acetylation and transcriptional activity. However, understanding of the mechanisms underlying this correlation remains incomplete. In addition to regulating PIC assembly and Pol2 initiation, our

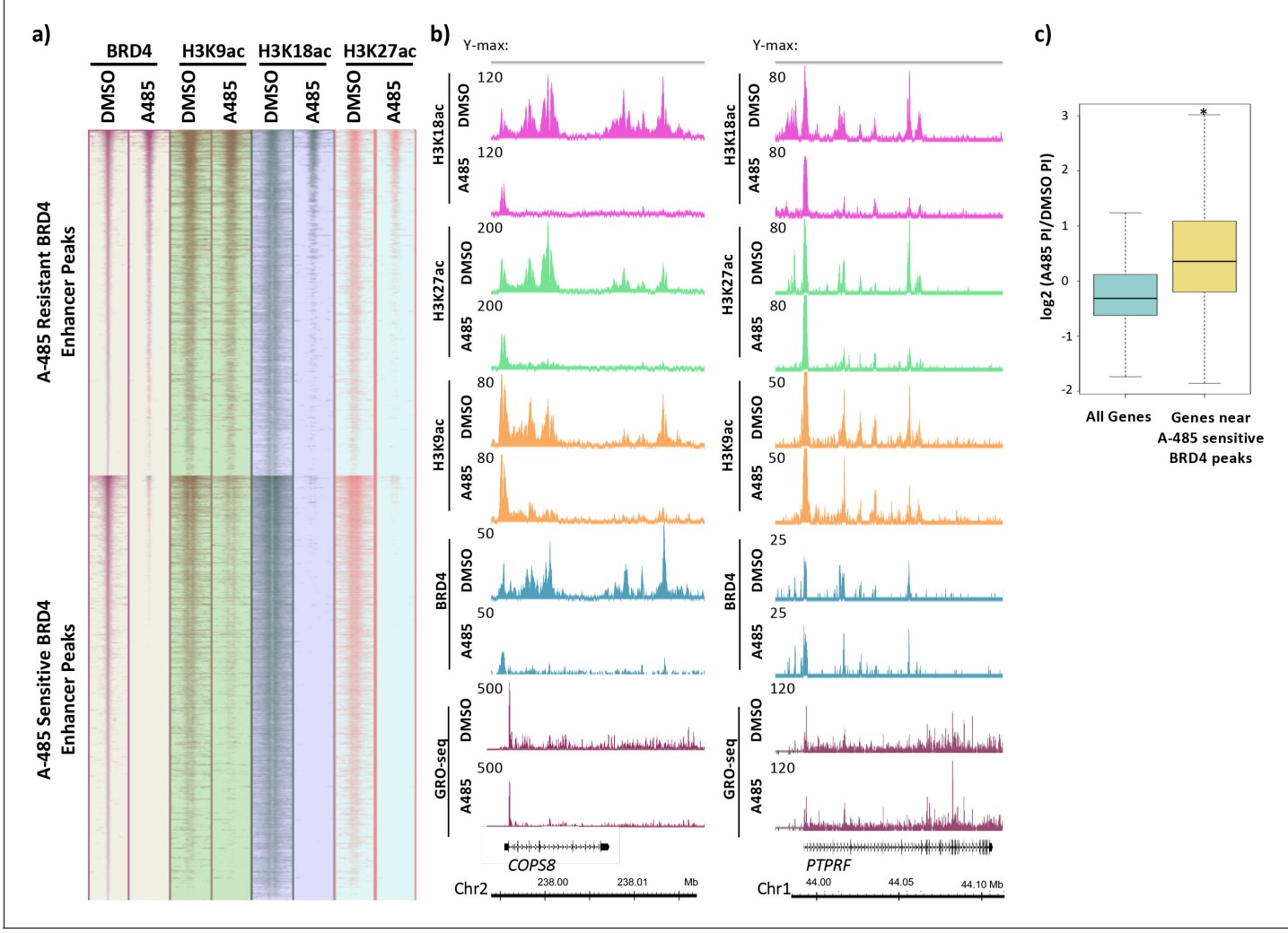

**Figure 7.** BRD4 enhancer binding stimulates pause release at nearby genes. (**a**) Heatmaps of BRD4, H3K9ac, H3K18ac, and H3K27ac ChIP-seq data. BRD4 enhancer peaks are divided into those that are A-485-resistant (top cluster) or A-485-sensitive (bottom cluster). (**b**) Gene browser plots of ChIP-seq data for the indicated histone modifications and BRD4, and GRO-seq counts for regions including the *COPS8* (left) and *PTPRF* (right) genes. (**c**) Boxplots comparing the change in pausing index (PI) after A-485 treatment (log2 (A-485 PI/DMSO PI)) for all genes versus genes near A-485-sensitive enhancer BRD4 peaks. *p<0.05, two-sided t-test.

The online version of this article includes the following figure supplement(s) for figure 7:

**Figure supplement 1.** Global Run-On sequencing counts plotted with low $y_{max}$ to show the relative level in the gene bodies of COPS8 and PTPRF in response to A-485.

**Figure supplement 2.** Global Run-On sequencing counts plotted with high and low $y_{max}$ to show the relative level of paused polymerase II near the transcription start site in A-485-treated and control DMSO-treated cells (observed in plot with $y_{max}$ = 2500) and in the gene body of the *NDRG1* gene (best observed at $y_{max}$ = 400).

**Figure supplement 3.** A-485 inhibited polymerase II (Pol2) release from the *PAG1* promoter-proximal pause site, but not Pol2 initiation and transcription to the *PAG1* pause site.

**Figure supplement 4.** A-485 inhibition of transcription initiation of *CSF3,* but not *PSMD3,* to the left also transcribed from the plus strand, or *MED24* to the right, transcribed from the minus strand.

results support a mechanism by which histone H3 N-terminal tail acetylation in the promoter-proximal region regulates Pol2 release from promoter-proximal pause sites in a subset (~4.3%) of active cellular promoters in primary human airway epithelial cells.

## H3K18/27 acetylation by CBP/p300 stimulates BRD4 and SEC association and promoter-proximal Pol2 pause release

Initially we observed that SEC recruitment through association with promoter region histone H3 acetylated at K18 and K27 stimulates paused Pol2 release at the HAdV-5 E4 promoter (*Figure 1a*, *Figure 2b*). To do this study, we used a multi-site E1A mutant (DM-E1A) defective for association of the large E1A AD (amino acids 121—225, *Figure 1—figure supplement 1*) with CBP/p300 (*Hsu et al., 2018*). GRO-seq studies following infection of human primary airway epithelial cells (HBTECs) with Ad5 vectors expressing wt or DM-E1A revealed that the E4 pausing index increased when the DM-E1A failed to stimulate H3K18/27ac at the E4 promoter. This result suggested that promoter H3K18/27ac contributes to paused Pol2 release at the E4 promoter.

We observed a correlation between defective paused Pol2 release at the viral E4 promoter as assayed by GRO-seq and decreased association of SEC subunits CDK9, AF9, and ENL assayed by ChIP-seq, indicating that H3K18/27ac is necessary for maximal SEC recruitment and Pol2 pause release in the E4 promoter-proximal region. This is similar to a proposed function of H3K9ac as a binding site for AF9 and ENL, thereby promoting paused Pol2 release by directly recruiting the SEC (*Gates et al., 2017*). SEC recruitment at E4 by H3K18/27ac may be due to interactions with acetyl-lysine-binding YEATS domains present in the AF9 and ENL SEC subunits (*Li et al., 2014*; *Gates et al., 2017*), but other SEC components may also contribute (*Gao et al., 2020*).

Defective pause release and decreased SEC recruitment at E4 also were observed in wt E1A-expressing cells when CBP/p300 acetyl transferase activity was inhibited by the highly specific competitive inhibitor A-485 (*Lasko et al., 2017*; *Figure 4a*). These results indicate that CBP/p300 HAT activity is necessary for maximal promoter-proximal paused Pol2 release at E4. BRD proteins (*BRD1-4)* have two BRD domains, at least one of which binds acetylated lysines with moderate affinity, potentially participating in cooperative protein binding to a region of chromatin with multiple acetylated lysines (*Jang et al., 2005*; *Dhalluin et al., 1999*; *Ozato et al., 2008*). Additionally, the mediator complex subunit MED26 is known to recruit the SEC after dissociation of the mediator from TFIID (*Takahashi et al., 2011*; *Vijayalingam and Chinnadurai, 2013*). Therefore, it is also possible that additional consequences of promoter-proximal H3K18/27 hypoacetylation, such as reduced association with MED26-containing mediator complexes, also contribute to decreased Pol2 promoter-proximal pause release at E4 after activation by DM-E1A.

The principal effect of E4 promoter H3K18/27 hypoacetylation on E4 transcription was on promoter-proximal pause release, causing a decrease in transcribing Pol2 downstream from the pause site, revealed by low GRO-seq counts in the gene body (*Figure 1a*). This correlated with lower SEC subunit association at the E4 TSS and reduced BRD4 throughout the E4 gene body after activation by DM-E1A compared to wt E1A (*Figure 2b*).

## A subset of cellular promoters requires H3K18/27 acetylation by CBP/ p300 for maximal promoter-proximal Pol2 release

Analysis of the ChIP-seq and GRO-seq data for cellular chromatin from cells treated with A-485 established that regulation of Pol2 pause release and SEC recruitment by promoter region H3K18/27ac also occurs at a small fraction of highly regulated cellular promoters. Following A-485 treatment, we observed the expected decreases in H3K18/27ac at typical enhancers and to an even greater extent at super-enhancers (*Figure 5a*). This accounts for the average total cell H3K18/27 hypoacetylation observed by western blotting of total cell protein (*Figure 3a*) since most H3K18/27 acetylation occurs in enhancers (*Figure 5—figure supplement 2b*). However, we observed a striking and unexpected *increase* in average H3K18/27ac at TSSs of all active genes, even in the presence of a concentration of A-485 sufficient to decrease total cell H3K18/27ac by >90% (*Figure 3a*, *Figure 5a*). These observations indicate that the dynamics of HAT and/or histone deacetylase activities differ in response to A-485 at promoters versus enhancers.

A-485 also resulted in defects in pause release assayed by GRO-seq at ~4.3% of active promoters in primary respiratory epithelial cells. These ~4.3% of promoters are similar to each other and different from most promoters in that promoter region acetylation was decreased after A-485 treatment as opposed to the increase in promoter H3K18/27ac observed for the average of all active genes (*Figure 6a*).

## Higher rate of H3K18/27 acetylation at promoters compared to enhancers

The steady-state level of H3K18/27ac on any specific nucleosome is determined by the relative rates of its acetylation and deacetylation (*Weinert et al., 2018*; *Shahbazian and Grunstein, 2007*). A-485 inhibits CBP/p300 acetyl transferase activity by competing with acetyl-CoA for binding to the enzyme's active site (*Lasko et al., 2017*). No evidence for inhibition of a histone deacetylase by A-485 was detected (*Lasko et al., 2017*) and is very unlikely given the highly specific interactions of A-485 with the CBP acetyl-CoA binding pocket (*Lasko et al., 2017*). Consequently, the decrease in average enhancer H3K18/27ac in A-485-treated cells (*Figure 5a*) suggests that the rate of H3K18/27 acetylation at enhancers in A-485-treated cells is greatly reduced compared to the normal rate in control DMSO-treated cells, as expected from the mechanism of a competitive inhibitor. In striking contrast to the expected decrease in H3K18/27ac at enhancers, at promoters the average H3K18/27ac *increased* during treatment with this specific inhibitor of the CBP/p300 acetyl transferase activity. This result suggests that H3K18/27ac is more stable at promoters than at enhancers and/or that the rate of H3K18/27 acetylation at promoters is higher than at enhancers. These results also suggest that there is an uncharacterized homeostatic mechanism that maintains promoter region H3K18/27ac in the face of extensive inhibition of the known lysine acetyl transferases that acetylate these sites, the closely related CBP and p300 (*Jin et al., 2011*).

It is possible that the difference in the effects of A-485 on the rates of promoter versus enhancer acetylation by CBP/p300 is due to differences in nucleosome density or the density of other proteins at promoters versus enhancers such that high protein concentration near promoters restricts the diffusion of the 536 Da A-485 drug molecule to the CBP/p300 active site. However, it seems unlikely that diffusion of A-485 molecules would be greatly restricted by nucleosomes that are ~400 times larger than the drug and irregularly packed into disordered chains of 'beads on a string' nucleosomes with variable particle and linker DNA arrangements in interphase nuclei (*Ou et al., 2017*; *Eagen et al., 2015*). Consequently, the resistance of promoters to A-485-induced hypoacetylation compared to enhancers in living cells probably is a result of a faster average rate of H3K18/27 acetylation by CBP/p300 at promoters than at enhancers. This is the result expected if transient interactions between the ADs of activators bound to their cognate DNA-binding sites in enhancers increase the local concentration of CBP/p300 in promoter regions.

BRD4 contains two bromodomains that bind acetylated lysines (*Filippakopoulos et al., 2012*; *Kanno et al., 2004*). The C-terminal portion of BRD4 binds P-TEFb and is thought to recruit it to hyperacetylated genomic regions to stimulate elongation (*Chen et al., 2018*). BRD4 has been shown to associate with promoters and enhancers and act as a histone chaperone to facilitate Pol2 elongation through chromatin to produce both protein-coding and enhancer RNAs (*Kanno et al., 2014*). By clustering BRD4 enhancer peaks into A-485-sensitive and A-485-resistant groups and correlating these peaks with H3K9ac and H3K18/27ac ChIP-seq data, we conclude that H3K9ac is sufficient for BRD4 recruitment at enhancers in the absence of H3K18/27ac (*Figure 7a*). Additionally, GRO-seq in cells treated with A-485 revealed a correlation between decreased BRD4 enhancer association and defective release of paused Pol2 from nearby promoters. The mechanism by which this occurs is likely through direct promoter–enhancer interactions facilitated by long-range chromatin interactions (*Takahashi et al., 2011*). Another possibility is that transcription of enhancer RNAs facilitated by BRD4 stimulates paused Pol2 release by promoting NELF release at a promoter brought into close proximity to the enhancer by folding of the chromatin fiber (*Schaukowitch et al., 2014*).

Our results suggest a model in which histone H3K18/27 acetylation by CBP/p300 in the promoter region is essential for maximal paused Pol2 release at the HAdV-5 E4 promoter. By analyzing H3 acetylation, SEC subunit chromatin association, and Pol2 pausing across the human genome, we established that regulation of Pol2 release from the NELF-induced pause site by H3K18/27ac also occurs at ~680 active human promoters in primary airway epithelial cells. Additionally, the identification of BRD4 enhancer peaks that were either sensitive or resistant to A-485 treatment presented an opportunity to study the effects of elongation factor association with enhancers on transcriptional elongation through neighboring transcription units. Interestingly, we found that BRD4 association with enhancers correlated with decreased Pol2 pausing and increased productive elongation from neighboring promoters.

A consensus TATA box (TATAWAWR) in the promoter regions of cellular genes correlated with increased pausing for 2XPI genes and decreased initiation for 2XIn genes in response to A-485 (*Figure 5—figure supplement 1*). TATA boxes are common in promoters of tightly regulated genes with focused TSSs, while TATA-less CpG island promoters are more common for constitutively expressed housekeeping genes with many unfocused TSSs (*Juven-Gershon and Kadonaga, 2010*). Since 2XPI and 2XIn genes are enriched for developmental functions with tightly regulated expression, enrichment for TATA boxes in the promoters of these genes is expected. This fits with the model that these tightly regulated genes are dependent on the dynamic enhancer H3K18/27ac for either transcription initiation, elongation, or both. Taken together, our results draw interesting causal links between histone H3 acetylation and regulation of Pol2 elongation as well as initiation.

# Materials and methods

## Key resources table

| Reagent type (species) or resource | Designation | Source or reference | Identifiers | Additional information |
|---|---|---|---|---|
| Cell line (human) | HBTEC (human bronchial /tracheal epithelial cells, primary) | Lifeline Cell Technology | FC-0035 | |
| Other | Ad5 mutant vector wt E1A | Constructed as previously described (*Hsu et al., 2018*) | | PMID:29976669 |
| Other | Ad5 mutant vector DM E1A | Constructed as previously described (*Hsu et al., 2018*) | | PMID:29976669 |
| Antibody | H3K18ac (rabbit polyclonal) | Prepared and validated as described previously (*Ferrari et al., 2012*) | | PMID:22499665 WB (1:2000) ChIP-seq (2 µL/ChIP) |
| Antibody | H3K9ac (rabbit polyclonal) | Millipore | 07-352 | WB (1:1000) ChIP-seq (2 µL/ChIP) |
| Antibody | H3K27ac (rabbit polyclonal) | Active Motif | 39133 | ChIP-seq (5 µL/ChIP) |
| Antibody | H3 (mouse monoclonal) | Abcam | ab10799 | WB (1:10,000) |
| Antibody | AF9 (rabbit polyclonal) | Genetex | GTX102835 | ChIP-seq (2 µL/ChIP) |
| Antibody | ENL/MLLT1 (rabbit monoclonal) | Cell Signaling | 14893 | ChIP-seq (10 µL/ChIP) |
| Antibody | BRD4 (rabbit polyclonal) | Bethyl | A301-985A50 | ChIP-seq (2 µL/ChIP) |
| Antibody | NELF TH1L (D5G6W) (rabbit monoclonal) | Cell Signaling | 12265S | ChIP-seq (7 µL/ChIP) |
| Antibody | Pol2 Ser2-P (31Z3G) (rabbit monoclonal) | Cell Signaling | 13499 | ChIP-seq (10 µL/ChIP) |
| Antibody | Pol2 Ser5-P (D9N5I) (rabbit monoclonal) | Cell Signaling | 13523 | ChIP-seq (10 µL/ChIP) |
| Antibody | CDK9 (C12F7) (rabbit monoclonal) | Cell Signaling | 2316 | ChIP-seq (10 µL/ChIP) |
| Antibody | BrdU antibody, agarose conjugated | Santa Cruz Biotechnology | sc-32323 AC | 50 µL/IP |
| Commercial assay, kit | TruSeq Small RNA Library Preparation Kit | Illumina | RS-200–0012 | |
| Commercial assay, kit | KAPA Hyper Prep kit | KAPA Biosystems | KK8504 | |
| Commercial assay, kit | NEXTflex ChIP-seq barcodes | Bio Scientific | NOVA-514120 | |
| Chemical compound, drug | A-485 | MedChemExpress | HY-107455 | |
| Software, algorithm | HISTAT2 | HISTAT2 | | PMID:31375807 |
| Software, algorithm | HTseq | HTseq | | PMID:25260700 |
| Software, algorithm | Homer | Homer | | PMID:20513432 |
| Software, algorithm | Bowtie2 | Bowtie2 | | PMID:22388286 |
| Software, algorithm | MACS2 | MACS2 | | PMID:18798982 |
| Software, algorithm | CEAS | CEAS | | PMID:19689956 |

## Ad5 mutant vectors

Ad5 mutant vectors expressing wt E1A and DM E1A were constructed as previously described (*Hsu et al., 2018*).

## Cell culture

HBTECs (catalog number FC-0035, lot number 02196; Lifeline Cell Technology) were grown at 37°C in a BronchiaLife medium complete kit (LL-0023; Lifeline Cell Technology) in a 5% $CO_2$ incubator until they reached confluence. Cells were then incubated 3 days more without addition of fresh medium and were infected for 12 hr with the indicated HAdV-5 mutants in the conditioned medium. A-485 (MedChemExpress) was added to a final concentration of 10 µM, or the same volume of DMSO (dimethyl sulfoxide) vehicle was added, and cells were incubated for an additional 2 hr.

## GRO-seq

Cells were harvested and incubated in swelling buffer (10 µM Tris-HCl, 2 mM $MgCl_2$, 3 mM $CaCl_2$). Nuclei were isolated with lysis buffer (10 µM Tris-HCl, 2 mM $MgCl_2$, 3 mM $CaCl_2$, 10% glycerol, 1% NP-40). Nuclear run-on was performed at 30°C for 7 min in 10 mM Tris-HCl pH 8, 5 mM $MgCl_2$, 300 mM KCl, 1 mM DTT, 500 µM ATP, 500 µM GTP, 500 µM Br-UTP, 2 µM CTP, 200 U/mL Superase In RNase Inhibitor (Invitrogen), and 1% sarkosyl. Nuclear RNA was isolated with Trizol (Invitrogen). DNAse treatment was performed with Turbo DNA-free kit (Invitrogen). RNA was purified with Micro Bio-Spin P-30 Gel Columns (Bio-Rad), fragmented with RNA Fragmentation Kit (Invitrogen), and treated with 10 units RppH (NEB) and 30 units T4 PNK (NEB). RNA immunoprecipitation was performed with anti-BrdU-conjugated agarose beads (Santa Cruz Biotechnologies). Library preparation was performed with TruSeq Small RNA Library Preparation Kit (Illumina). GRO-seq reads were aligned with HISTAT2 software to Ad5 and human (hg19) genomes and normalized to the number of reads aligned to hg19. Pause indexes (TSS to +200 counts)/(+201 to TTS counts) were calculated using HTSeq software. Genes determined to be affected by A-485 underwent additional profiling using Homer software (http://homer.ucsd.edu/homer/; *Heinz et al., 2010*) for biological processes gene ontology enrichment as well as TF motif analysis in promoter regions (± 300 bp from TSSs).

## qRT-PCR

Total RNA extracted from HTBECs using a PureLink RNA minikit (Ambion) was reverse transcribed with random hexamer priming using Superscript III (Invitrogen). RNA was treated with DNase I with Turbo DNA-free kit (Ambion). Quantitative reverse transcription-PCRs (qRT-PCRs) were carried out with the Applied Biosystems 7500 real-time PCR system with FastStart universal SYBR green master mix (Roche). All values were normalized to 18S RNA levels.

## ChIP-seq

Preparation of cross-linked HBTEC chromatin, sonication, and immunoprecipitation was as described in *Ferrari et al., 2014*. Sequencing libraries were constructed from 1 ng of immunoprecipitated and input DNA using the KAPA Hyper Prep kit (KAPA Biosystems) and NEXTflex ChIP-seq barcodes (Bio Scientific).

## Data analysis of ChIP-seq

ChIP-seq libraries were sequenced using HiSeq 4000 or NovaSeq 6000. For analysis on the Ad5 genome, sequence tags were aligned using Bowtie2 software and normalized to the following formula: (number of Ad5-aligned reads in the input sample/number of human-aligned reads in the input sample) × (number of Ad5-aligned reads in the ChIP sample). For analysis on the human genome, reads were mapped to the hg19 human genome reference using Bowtie2 software. Only reads that aligned to a unique position in the genome with no more than two sequence mismatches were retained for further analysis. Duplicate reads that mapped to the same exact location in the genome were counted only once to reduce clonal amplification effects. MACS2 software was used for peak calling (q-value <0.05 was considered significant). The total counts of the input and ChIP samples were normalized to each other. Samples were normalized for equal number of uniquely mapped reads. The input sample was used to estimate the expected counts in a window. Wiggle files were generated using a custom algorithm and present the data as normalized tag density as seen in all

figures with genome browser shots. Metagene plots displaying normalized average relative ChIP-seq signals were generated using CEAS software.

## Antibodies

Antibodies included H3K18ac (814), prepared and validated as described previously (*Ferrari et al., 2012*), H3K9ac (07-352; Millipore), H3K27ac (39133; Active Motif), H3 (ab10799, Abcam), AF9 (GTX102835, Genetex), ENL/MLLT1 (14893, Cell Signaling), BRD4 (A301-985A50), NELF TH1L D5G6W (12265S, Cell Signaling), Pol2 Ser2-P 31Z3G (13499, Cell Signaling), Pol2 Ser5-P D9N5I (13523, Cell Signaling), and CDK9 C12F7 (2316, Cell Signaling).

# Additional information

## Funding

| Funder | Author |
| --- | --- |
| Professor June Lascelle Fund | Arnold J Berk |

The funders had no role in study design, data collection and interpretation, or the decision to submit the work for publication.

## Author contributions

Emily Hsu, Conceptualization, Data curation, Formal analysis, Writing - original draft; Nathan R Zemke, Formal analysis, Writing - review and editing; Arnold J Berk, Conceptualization, Funding acquisition, Investigation, Writing - original draft

## Author ORCIDs

Emily Hsu (iD) https://orcid.org/0000-0001-5096-0745
Nathan R Zemke (iD) https://orcid.org/0000-0002-6326-5925
Arnold J Berk (iD) https://orcid.org/0000-0001-9379-6287

## Decision letter and Author response

Decision letter https://doi.org/10.7554/eLife.63512.sa1
Author response https://doi.org/10.7554/eLife.63512.sa2

# Additional files

## Supplementary files

• Transparent reporting form

## Data availability

Sequencing data have been deposited in GEO under accession code GSE167094.

The following dataset was generated:

| Author(s) | Year | Dataset title | Dataset URL | Database and Identifier |
| --- | --- | --- | --- | --- |
| Berk AJ | 2021 | Promoter-specific changes in initiation, elongation and homeostasis of histone H3 acetylation during CBP/p300 Inhibition | https://www.ncbi.nlm.nih.gov/geo/query/acc.cgi?acc=GSE167094 | NCBI Gene Expression Omnibus, GSE167094 |

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
