## [Decision Letter]

**Acceptance summary:**

This study addresses the role of histone acetylation on the release of paused RNA polymerase II from Adenovirus early promoters and from cellular promoters. By combining measurement of nascent polymerase II generated RNA and chromatin immunoprecipitation for RNA polymerase II, BRD4 and components of the super elongation complex in the presence or absence of A-485 a P300/CBP inhibitor, the authors identify both Adenoviral and cellular promoters where P300/CBP inhibition affects RNA polymerase II initiation or more specifically release of RNA polymerase II pausing. The results also revealed that P300/CBP inhibition has differential effects on histone acetylation at super enhancers, standard enhancers and proximal promoters suggesting a novel homeostatic mechanism that regulates acetylation and/or its turnover at these different regions. The study specifically uncovers a group cellular promoters that are particularly sensitive to P300/CBP inhibition leading to decreased released of paused RA polymerase II. This is a well performed study that will be of general interest to researchers investigating the mechanisms regulating gene expression, in particular how chromatin acetylation regulates release of RNA polymerase II pausing, a major regulatory step in transcription.

**Decision letter after peer review:**

Thank you for submitting your article "Promoter-specific changes in initiation, elongation and homeostasis of histone H3 acetylation during CBP/p300 inhibition" for consideration by *eLife*. Your article has been reviewed by two peer reviewers, including Irwin Davidson as the Reviewing Editor and Reviewer #1, and the evaluation has been overseen by Kevin Struhl as the Senior Editor.

The reviewers have discussed the reviews with one another and the Reviewing Editor has drafted this decision to help you prepare a revised submission.

Summary:

This study addresses the role of histone acetylation on the release of paused Pol II first from Adenovirus early promoters and then extended to all cellular promoters. By combining Gro-Seq to assess Pol II nascent transcription and ChIP-seq for modified Pol II, BRD4 and SEC components in the presence or absence of A-485 a P300/CBP inhibitor, the authors identify both Adenoviral and cellular promoters where P300/CBP inhibition affects Pol II initiation or more specifically release of Pol II pausing. The study specifically uncovers a group of around 1000 cellular promoters that are particularly sensitive to P300/CBP inhibition leading to decreased released of paused Pol II. The results also indicate that P300/CBP inhibition has different effects on histone acetylation at enhancers and proximal promoters/TSS.

The reviewers think this is a well-conducted study that yields important novel insights in the connection between histone acetylation by the CBP/p300 coactivators and Pol II pause release by the SEC. This manuscript is well suited to the general readership of e*Life*.

However, before acceptance, several issues should be addressed in a revised version.

Essential revisions:

1) The effect of A-485 on ENL recruitment to the E4 promoter in Figure 4A are minimal and much weaker than on AF9 binding. The authors should strengthen their case by including a CDK9 ChIPseq on A-485 treated Ad-infected cells.

2) In the subsection “Decreased Pol2 pause-release in the E4 promoter-proximal region correlates with decreased association of SEC subunits CDK9, AF9, and ENL”, the authors speculate that decreases in pol II elongation would lead to increased phosphor-CTD signals. To provide data for this, the authors could perform a ChIPseq on a core pol II subunit.

3) The authors should provide a statistical test of the π differences mentioned in the subsection “CBP/p300 acetyl-transferase activity is required for efficient Pol2 pause-release and recruitment of AF9, ENL, and BRD4 at E4”.

4) In the context of the Adenovirus early promoters, the authors discuss that the presence of a consensus TATA-element can account for resistance to histone hypo-acetylation, but they do not extend this analyses to the cellular genes. Have they investigated the relationship between consensus TATA-elements and sensitivity of Pol II initiation/pause release at cellular genes to inhibition by A-485, in particular as it has been shown that only a limited subset of cellular promoters comprises consensus TATA-elements?

5) When analyzing the cellular promoters and their differential responses to A-485, there is no investigation of “super-enhancers” that are characterized by extended regions of histone acetylation and BRD4 occupancy, key actors in this study. In Figure 7—figure supplement 4, there seems to be a “super-enhancer” region where histone acetylation is strongly affected. Can the authors comment on whether “super-enhancers” are more affected than other enhancers by A-485 and what are the effects on transcription of the associated genes.

6) While the study identifies cellular genes where release from Pol II pausing is modulated by histone acetylation, there is no investigation of the underlying mechanism that would explain the differential sensitivity of these 2XPI promoters. Why is H3 acetylation lowered at these promoters but increased at the majority of the others? For example, have they investigated the sequence motifs seen at these promoters or associated enhancers to assess whether binding motifs for specific factors are enriched at promoters where Pol II release is sensitive/resistant to acetylation levels and recruitment of SEC components, have they assessed a relationship to GC content and/or position of the +1 nucleosomes? It is important that the authors provide some data or at least some discussion of which characteristics set these promoters apart and would account for their differential sensitivity to A-485 treatment and release of paused Pol II.

---

## [Author Response]

Essential revisions:1) The effect of A-485 on ENL recruitment to the E4 promoter in Figure 4A are minimal and much weaker than on AF9 binding. The authors should strengthen their case by including a CDK9 ChIPseq on A-485 treated Ad-infected cells.

We would prefer to not pursue an explanation for the difference between the magnitude of the effect of A-485 on AF9 versus ENL association with the Ad5 genome. This is because the principal importance of the ChIP-seq results for AF9 and ENL on the Ad5 genome was that they led us to do the analysis of AF9 and ENL association with cellular genes shown in Figure 6B. We observed an unequivocal and much more extensive decrease in both AF9 and ENL ChIP-seq reads on cellular DNA after A-485 treatment than on the Ad5 genome. This may be because the increase in π caused by A-485 was much greater on the 2XPI cellular genes (≥ 2 fold) than on the Ad5 E4 region (1.4 fold).

In the revised manuscript, in order to be clear about the relatively small effect of A-485 we now present quantitation based on the total ChIP-seq read counts under the peaks in the ENL ChIP-seq showing that the ENL read counts were reduced only ~20% by A-485 treatment, a relatively small effect as the reviewers note.

New text: "To determine if inhibition of CBP/p300 HAT activity resulted in defective SEC recruitment at E4, we performed AF9, ENL, and BRD4 ChIP-seq in cells infected with the wt E1A vector after DMSO or A-485 treatment. As with cells expressing DM-E1A, A-485 treatment greatly decreased AF9 association at the E2early, E3, and E4 promoter regions, and decreased the number of counts in ENL peaks from the E2early TTS to the right end of the Ad5 genome from 13,712,875 to 10,804,279, an ~20% decrease (Figure 4A)."

2) In the subsection “Decreased Pol2 pause-release in the E4 promoter-proximal region correlates with decreased association of SEC subunits CDK9, AF9, and ENL”, the authors speculate that decreases in pol II elongation would lead to increased phosphor-CTD signals. To provide data for this, the authors could perform a ChIPseq on a core pol II subunit.

We did not mean to “speculate that decreases in pol II elongation would lead to increased phosphorCTD signals.” In this portion of the submitted manuscript we were discussing the small peaks of Pol2 Ser2-P and Pol2 Ser5-P observed just upstream of the short E2 second exon in Figure 2A. These peaks were not a major subject of this paper. The relevant passage reads:

“We observed two Ser2-P peaks in the E2early gene body, one just downstream of the TSS, likely indicating paused Pol2. Another Ser2-P peak occurred over the E2early second exon. […] Such a decrease in Pol2 elongation rate over the short E2 second exon would cause an increase in the steady-state level of Pol2 over the exon, potentially leading to the increase in the Pol2 ChIP-seq signal observed over the E2 second exon.”

Contrary to the statement in the Review point (2), we did not “speculate that decreases in pol II elongation would lead to increased phosphorCTD signals.” We were simply providing a potential explanation for the small Ser2-P and Ser5-P peaks over the E2 second exon, which were not important points of the paper, and were not discussed further.

To clarify our point here and avoid confusion, we changed the wording to:

“These Pol2 peaks just upstream of the E2 second exon may be explained by a previously reported reduction in elongation rate over some exons, proposed to influence splice site recognition and spliceosome assembly (Jonkers, Kwak and Lis, 2014; Martin et al., 2013).”

3) The authors should provide a statistical test of the π differences mentioned in the subsection “CBP/p300 acetyl-transferase activity is required for efficient Pol2 pause-release and recruitment of AF9, ENL, and BRD4 at E4”.

We have now performed three independent biological replicates of the GRO-seq assays on the Ad5 genome in HBTECs infected with the wt E1A and DM E1A vectors. From this data we calculated the fold change in pause index (PI) at the viral early promoters between wt E1A which makes the E1A-AD interaction with p300/CBP and DM E1A which is blocked specifically for the p300/CBP interaction (as opposed to several other E1A interactions with other host proteins Figure 1—figure supplement 1). We calculated the average change in π and the statistical significance (p-value) of the differences between wt E1A and DM-E1A. The p-values are now shown in the new Figure 1B, and the figure legend has been modified accordingly:

“(B) Average fold change in pause index for E2early, E3, and E4 in cells expressing DM-E1A compared to wt E1A. Pause index is the ratio of reads in the promoter region (TSS to +200) to reads in the gene body (+200 to TTS). […] * indicates a significant difference (p-value < 0.05) between cells expressing wt E1A and DM-E1A. “ns” indicates no significant difference.”

4) In the context of the Adenovirus early promoters, the authors discuss that the presence of a consensus TATA-element can account for resistance to histone hypo-acetylation, but they do not extend this analyses to the cellular genes. Have they investigated the relationship between consensus TATA-elements and sensitivity of Pol II initiation/pause release at cellular genes to inhibition by A-485, in particular as it has been shown that only a limited subset of cellular promoters comprises consensus TATA-elements?

We are grateful to the reviewers for asking us to clarify this point. The new Figure 5—figure supplement 1 shows that ~4.5% of 2XPI promoters and ~4.8% of 2XInitiation promoters have consensus TATA-boxes (TATAWAAR), compared to ~1.8% of all active promoters.

5) When analyzing the cellular promoters and their differential responses to A-485, there is no investigation of “super-enhancers” that are characterized by extended regions of histone acetylation and BRD4 occupancy, key actors in this study. In Figure 7—supplement 4, there seems to be a “super-enhancer” region where histone acetylation is strongly affected. Can the authors comment on whether “super-enhancers” are more affected than other enhancers by A-485 and what are the effects on transcription of the associated genes.

Again, we are grateful to the reviewers for asking us to clarify this point. The new Figure 5—figure supplement 2 shows the total number of peaks and the percentage of total peaks of H3K27ac, H3K18ac, H3K9ac, and BRD4 in super-enhancers, typical enhancers, and promoters in primary human tracheal epithelial cells and the changes in these values in A-485 treated and control DMSO vehicle treated cells. As the reviewers suggested, these plots and numbers of peaks show that “super-enhancers” are more affected than other enhancers by A-485. This is discussed in the revised manuscript:

“Comparing the average signals for H3K18ac and H3K27ac at all TSSs and enhancer peaks (defined as peaks >2.5kb from the nearest TSS), we observed the expected decreases in H3K18ac and H3K27ac by A-485 at enhancer peaks (Figure 5A). […] Furthermore, the percentage of total H3K27ac and H3K18ac peaks increased at promoters and decreased at enhancers after A-485 treatment, while the effects on H3K9ac peaks were minimal (Figure 5—figure supplement 2).”